# Long Non-Coding RNAs in Hypoxia and Oxidative Stress: Novel Insights Investigating a Piglet Model of Perinatal Asphyxia

**DOI:** 10.3390/biology12040549

**Published:** 2023-04-04

**Authors:** Benedicte Grebstad Tune, Maria Melheim, Monica Åsegg-Atneosen, Baukje Dotinga, Ola Didrik Saugstad, Rønnaug Solberg, Lars Oliver Baumbusch

**Affiliations:** 1Department of Pediatric Research, Division of Paediatric and Adolescent Medicine, Oslo University Hospital Rikshospitalet, 0372 Oslo, Norway; b.g.tune@studmed.uio.no (B.G.T.);; 2Department of Health, Nutrition and Management, Oslo Metropolitan University, 0130 Oslo, Norway; 3Institute of Clinical Medicine, University of Oslo, 0450 Oslo, Norway; 4Department of Pediatrics, Division of Neonatology, Beatrix Children’s Hospital, University Medical Center Groningen, University of Groningen, 9713 GZ Groningen, The Netherlands; 5Department of Pediatrics, Vestfold Hospital Trust, 3103 Tønsberg, Norway; 6Faculty of Health, Welfare and Organization, Østfold University College, 1757 Halden, Norway

**Keywords:** *BDNF-AS*, birth asphyxia, brain, hyperoxia, hypoxia, long non-coding RNA, oxidative stress, piglet model, perinatal asphyxia

## Abstract

**Simple Summary:**

Birth asphyxia (defined as the failure to establish breathing at birth) causes 800,000 newborn deaths worldwide every year and remains a leading cause of lifelong disability in children under five. Reliable diagnostic markers and advanced therapeutic approaches are still lacking. The aim of this study was to explore selected long non-coding RNAs (lncRNAs) and their protein-coding target genes using a piglet model of perinatal asphyxia. A total of 42 newborn piglets were randomized into 4 study groups. Expression of the lncRNAs (*BDNF-AS*, *H19, MALAT1, ANRIL*, *TUG1*, and *PANDA*) and their related protein-coding genes (*VEGFA*, *BDNF*, *TP53*, *HIF1α*, and *TNFα*) were quantified in the cortex, the hippocampus, the white matter, and the cerebellum. Our results indicate significant changes in lncRNA expression, mainly in the cortex and hippocampus. Furthermore, a brief exposure to 100% oxygen for 3 min is sufficient to induce alterations. Our observations suggest that lncRNAs are part of the molecular response to hypoxia-induced damage during perinatal asphyxia. A better understanding of the regulatory properties of lncRNAs may provide novel targets and intervention strategies for perinatal asphyxia in the future.

**Abstract:**

Birth asphyxia is the leading cause of death and disability in young children worldwide. Long non-coding RNAs (lncRNAs) may provide novel targets and intervention strategies due to their regulatory potential, as demonstrated in various diseases and conditions. We investigated cardinal lncRNAs involved in oxidative stress, hypoxia, apoptosis, and DNA damage using a piglet model of perinatal asphyxia. A total of 42 newborn piglets were randomized into 4 study arms: (1) hypoxia–normoxic reoxygenation, (2) hypoxia–3 min of hyperoxic reoxygenation, (3) hypoxia–30 min of hyperoxic reoxygenation, and (4) sham-operated controls. The expression of lncRNAs *BDNF-AS, H19, MALAT1, ANRIL, TUG1,* and *PANDA*, together with the related target genes *VEGFA*, *BDNF*, *TP53*, *HIF1α*, and *TNFα*, was assessed in the cortex, the hippocampus, the white matter, and the cerebellum using qPCR and Droplet Digital PCR. Exposure to hypoxia–reoxygenation significantly altered the transcription levels of *BDNF-AS, H19, MALAT1,* and *ANRIL. BDNF-AS* levels were significantly enhanced after both hypoxia and subsequent hyperoxic reoxygenation, 8% and 100% O_2_, respectively. Our observations suggest an emerging role for lncRNAs as part of the molecular response to hypoxia-induced damages during perinatal asphyxia. A better understanding of the regulatory properties of *BDNF-AS* and other lncRNAs may reveal novel targets and intervention strategies in the future.

## 1. Introduction

Birth asphyxia (intrapartum-related events) is the cause of approximately 800,000 newborn casualties annually and remains a leading cause of preventable deaths and lifelong impairment in children under the age of five years worldwide [1,2,3]. The condition is due to decreased oxygen supply or reduced blood flow to various organs, causing potential cellular and organ damage. Mild, moderate, or severe brain injuries, including hypoxic–ischemic encephalopathy (HIE), can lead to permanent complications and disabilities, such as cerebral palsy, epilepsy, as well as motor and learning impairments [4,5]. Despite the dramatic consequences, there is still a lack of preventive measures and good diagnostic markers to predict the severity of damage and therapeutic approaches are limited. One of the few possible beneficial applications for asphyxiated newborn infants is the use of supplemental oxygen. However, both hypoxia and hyperoxia (excess oxygen supply) may lead to the increased production of reactive oxygen species, resulting in oxidative stress. Consequently, a well-balanced level of oxygen is essential to avoid irreversible damage to organs and cellular structures [6,7]. The brain of newborns is a vulnerable target for irreversible injury caused by hypoxia and oxidative stress. Asphyxiated and preterm infants are especially susceptible due to their impaired or immature oxygenation regulation mechanisms [8].

The mammalian brain consists of specialized components, including the cortex, the hippocampus, the cerebellum, and the white matter. These disparate sections reveal structural, functional, and morphological differences and, accordingly, diverse characteristics in their susceptibility to exo- or endo-genic damages. Cells have developed diverse strategies to survive at low oxygen levels [9]. The hypoxia-inducible factor 1 alpha (*HIF1α*) is one of the master regulatory elements for the hypoxic response and triggers several downstream target genes [10], including the vascular endothelial growth factor A (*VEGFA*), the brain-derived neurotrophic factor (*BDNF*), the tumor suppressor gene p53 (*TP53*), and the tumor necrosis factor α (*TNFα*) [11].

In the last decade, it has become more and more obvious that the expression of protein-coding genes is a highly regulated process involving the interaction of cis- and trans-localized sequences, transcription factors, and other regulatory molecules, such as various types of non-coding RNAs. A selection of small non-coding RNA molecules, called microRNAs (miRNA), has previously been studied in relation to birth asphyxia [12]; however, the large and heterogeneous group of long non-coding RNAs (lncRNAs) has attracted less attention, despite the growing awareness that lncRNAs are involved in a wide range of developmental processes and conditions, including apoptosis, inflammatory response, and angiogenesis [13] (Table 1). 

Several lncRNAs are key regulators for the up- or downregulation of gene expression in response to hypoxia and oxidative stress, e.g., hypoxia-associated lncRNAs (HALs) are regulated by hypoxia-inducible factors (HIFs) via hypoxia response elements in their promoters [10,13,29,30,31]. HALs, along with other oxidative-stress-associated lncRNAs, are expressed in the developing and adult CNS, as well as during pathological brain injury, and hence, may influence the outcome by altering their expression [14,32,33]. 

The rapidly increasing number of publications on the impact of lncRNAs in various conditions and diseases suggests that universal lncRNAs, such as *HOTAIR* and *PANDA,* or disease-specific lncRNAs, are relevant for the observed damages in birth asphyxia [13,29,30,31]. Therefore, we decided to investigate a selection of both, global and hypoxia-specific lncRNAs, including brain-derived neurotrophic factor antisense (*BDNF-AS*), H19 imprinted maternally expressed transcript (*H19*), metastasis-associated lung adenocarcinoma transcript 1 (*MALAT1*), antisense non-coding RNA in the INK4 locus (*ANRIL*), taurine-upregulated gene 1 (*TUG1*), and p21-associated ncRNA DNA-damage-activated (*PANDA*) (Table 1). 

The overall aim of this study was to explore the changes in the expression of lncRNAs involved in the regulation of the oxidative stress response in perinatal asphyxia using a piglet model. Our results from different brain regions exhibited that exposure to hypoxia caused significant alterations in the expression of various lncRNAs, mainly in the cortex and hippocampus. Even a short exposure to 100% oxygen for 3 min caused expression changes in the lncRNAs and a significant increase in *BDNF-AS* in a time-dependent manner, providing further evidence for oxidative brain injury caused by hyperoxic reoxygenation in newborns. 

## 2. Material and Methods

### 2.1. Study Design

Newborn Noroc pigs (*n* = 42) were exposed to a well-developed piglet model of perinatal asphyxia at our facilities [34]. Piglets in good general condition between 12 and 36 h of age and with a weight of 1.8–2.2 kg were included in the study. The rectal temperatures of the piglets were continuously monitored and maintained (38.5–39.5 °C). Anesthesia was induced with a dose of fentanyl (0.05 mg/kg) and pentobarbital (15 mg/kg) intravenously provided through a cannula in an ear vein. The piglets were orally intubated, ventilated, and surgically prepared as described by Benterud et al. [35]. An infusion of fentanyl (0.05 mg/kg/h) and Benelyte^®^ was administered continuously throughout the procedure. Additional doses of pentobarbital were given if necessary (2.5 mg/kg). The piglets were stabilized for one hour before proceeding with the experimental protocol. After stabilization, the piglets were block-randomized into four different study arms: hypoxia (8% O_2_) and normoxic reoxygenation (21% O_2_); hypoxia and 3 min of hyperoxic reoxygenation (100% O_2_); hypoxia and 30 min of hyperoxic reoxygenation; and a sham-operated control group (Figure 1). We will continue to refer to these groups as Reox-21%, Reox-100% (3′), Reox-100% (30′), and control group. The animals included in the control group underwent the same procedures and observation times but were not exposed to hypoxia or hyperoxia.

Hypoxia was induced by ventilation with 8% O_2_ in N_2_ until the base excess (BE) reached −20 mM/L, or the mean arterial blood pressure (MABP) decreased to 20 mmHg before the piglets were reoxygenated (Appendix A). The Reox-21% group was purely reoxygenated with 21% O_2_. The hyperoxic groups were reoxygenated with 100% O_2_ for either 3 (Reox-100% (3′)) or 30 (Reox-100% (30′)) minutes, followed by 21% O_2_ for the remaining time of the experiment. The control group received 21% O_2_ throughout the entire experiment and was treated identically to the other groups but was not exposed to hypoxia or hyperoxia (Figure 1). Arterial blood gas measurements, heart rate, oxygen saturation, blood pressure, and temperature were continuously monitored. In addition, the anesthesia and analgesia of the pigs were continuously assessed. All the piglets were observed for 9.5 h after the end of hypoxia and then euthanized with an overdose of pentobarbital (150 mg/kg) intravenously. The brain was immediately removed, and tissue specimens were collected from the right hemisphere of the brain, including the prefrontal cortex, the hippocampus, the white matter, and the cerebellum. All tissue samples were snap-frozen in liquid nitrogen and stored at −80 °C until further analysis. Four piglets died before the completion of the experiment and were therefore excluded. An autopsy revealed organ abnormalities in three piglets from the control group, indicating that they were unsuitable as controls and were hence excluded (Appendix A).

### 2.2. RNA Isolation

The total RNA was isolated from various brain regions using an E.Z.N.A.^®^ Total RNA kit II–Animal Tissue Protocol (Omega Bio-Tek, Inc., Norcross, GA, USA) according to the manufacturer’s instructions (v4.0), with the following modifications: 2-Mercaptoethanol was not added to the RNA Solve reagent, and DNase I digestion was added to optimize DNA removal. 

The estimated RNA purity and concentrations were measured using a Nanodrop^®^ ND-1000 Spectrophotometer (NanoDrop Technologies, Inc., Wilmington, NC, USA). RNA integrity was measured using an RNA ScreenTape assay (Agilent, Santa Clara, CA, USA) on the Agilent 4200 TapeStation system according to the manufacturer’s instructions (September 2015). 

### 2.3. cDNA Synthesis

RNA was transcribed into cDNA using a high-capacity cDNA reverse-transcription kit (Applied Biosystems, a brand of Thermo Fisher Scientific Inc., Waltham, MA, USA) according to the manufacturer’s instructions (June 2010) using universal PCR settings on a PTC-100™ Programmable Thermo Controller (MJ Research, Inc., St. Bruno, QC, Canada) cDNA was diluted in nuclease-free water (NFW) to 10 ng/μL before performing gene expression analysis using qPCR. 

### 2.4. Quantitative Real-Time Polymerase Chain Reaction

Primers were designed using the Primer Express 3.0.1 software (Applied Biosystems) and purchased from Life Technologies AS (Table 2). Briefly, 20 µL reactions containing 40 ng cDNA, 400 nM primers, and 1 × Power SYBR Green (Applied Biosystems) were mixed in a MicroAmp^®^ Optical 96-Well Reaction Plate (Applied Biosystems), sealed with AB-1170 Optically Clear Adhesive Seal Sheets (Thermo Fisher Scientific Inc., Waltham, MA, USA), and analyzed on a Life Technologies ViiA 7 (Applied Biosystems) using universal settings. Melting curve analysis was performed to confirm the presence of a single target in the reaction.

Gene expression studies were performed using ViiA 7 RUO Software (Thermo Fisher Scientific), and absolute gene expression was calculated in Microsoft Excel (Microsoft, Redmond, WA, USA) using normalized Ct-values and the 2^−∆Ct^ method before performing statistical analysis. 

### 2.5. Digital Droplet PCR

Briefly, a 20 µL reaction mix containing 1 × QX200 ddPCR EvaGreen Supermix, 200 nM forward- and reverse primers, 20 ng cDNA template, and 70 µL of Droplet Generation Oil for EvaGreen^®^ were transferred to separate sample- and oil wells in a DG8™ Cartridge for QX200™ Droplet Generator (BioRad, Hercules, CA, USA), before sealing with the Droplet Generator DG8™ Gasket (BioRad). The samples were converted to droplets using the QX200 Droplet Generator (BioRad). Approximately 40 µL droplets were transferred to 96-Well ddPCR Plates, Semi skirted (BioRad), and heat-sealed with Pierceable Foil Heat Seal (BioRad) in a PX1 plate sealer at 180 °C for 5 s before thermal cycling (Appendix A) according to the manufacturer’s description on the Veriti™ 96-well Thermal Cycler (Applied Biosystems). The PCR lid was heated to 105 °C, and the sample volume was set to 40 µL. The 96-well plate was analyzed on a QX200 Droplet Reader using the QuantaSoft software (Thermo Fisher Scientific). 

Values from ddPCR were obtained from QuantaSoft software, and relative normalized expression was calculated in Microsoft Excel by applying Equations (1) and (2) before performing statistical analysis.
(1)$$Relative\;Quantity\;\left(RQ\right)=\frac{Treatment\;group\;\left(\mathrm{copies}/\mathrm{ul}\right)}{Control\;group\;\left(\mathrm{copies}/\mathrm{ul}\right)}$$
(2)$$Normalized\;expression\;\left(NE\right)=\frac{RQ\;of\;Treatment\;group}{RQ\;of\;reference\;gene}$$

### 2.6. Statistical Analysis

Statistical analyses were performed using GraphPad Prism 8 (GraphPad Prism Software Inc., Boston, MA, USA). Normal distribution was assessed using the Shapiro–Wilk normality test, and QQ plots were evaluated before further analysis of the data. If the criteria for normal distribution were not met, the data were log2-transformed to achieve normal distribution. If the normal distribution was still not met, a nonparametric test, Mann–Whitney test, or Kruskal–Wallis test, was applied to nontransformed data and expressed as mean ± interquartile range (IQR). An unpaired *t*-test or analysis of variance (ANOVA) was applied to normally distributed data and expressed as mean ± standard deviation (SD). Results with *p* < 0.05 were accepted as statistically significant. 

## 3. Results

Changes in the mRNA expression levels of the oxidative-stress-related protein-coding genes *VEGFA*, *BDNF, TP53*, *HIF1α*, and *TNFα* were measured using qPCR and compared with the quantities of the lncRNAs *BDNF-AS*, *H19*, *MALAT1*, *ANRIL*, *TUG1*, and *PANDA* in the Reox-21% versus the control group, in samples from the cortex, the hippocampus, the white matter, and the cerebellum generated using our piglet model, as defined in Figure 1 (Appendix A). In addition, the selected lncRNAs were analyzed using ddPCR in tissue from the cortex of all three intervention groups versus the control group (Appendix A provide a complete overview of the genes and brain regions analyzed in the four study arms using qPCR or ddPCR).

### 3.1. Differences in the mRNA Expression of Target Genes Measured in Various Brain Regions and after Exposure to Hypoxia and Normoxic Reoxygenation 

Analysis using qPCR revealed that the mRNA levels of *VEGFA* in the Reox-21% group were significantly upregulated in the cortex, hippocampus, and cerebellum samples compared with the control group. A similar trend, although not significant, was observed in the white matter (*p* = 0.0514) (Figure 2A). *BDNF* expression showed a tendency to increase in the cortex, the hippocampus (*p* = 0.0939), and the white matter (*p* = 0.3357); however, the upregulation was significant only in the cortex (*p* = 0.014). In the cerebellum, *BDNF* expression was approximately equal between the Reox-21% and the control groups (Figure 2A). Moreover, *HIF1α* mRNA quantities were similarly expressed in both groups and across samples from the cortex, the hippocampus, and the white matter. The *HIF1α* expression was elevated in the cerebellum compared with the other brain regions (Appendix A). *TP53* expression was not significantly up- or downregulated in any of the brain regions. *TNFα* was found to be significantly upregulated only in the cerebellum samples. We further observed a high intra-group variability for all the groups (Appendix A).

### 3.2. LncRNAs Are Significantly Altered in Various Brain Regions after Exposure to Hypoxia and Normoxic Reoxygenation

In the cortex samples, the gene expression of *BDNF-AS* (Figure 2B) analyzed using qPCR was upregulated in the Reox-21% group compared with the control group. Although a similar trend was observed in samples of the cerebellum (*p* = 0.0538), the difference was determined as nonsignificant. *BDNF-AS* gene expression in the hippocampus and white matter samples was not altered compared with the control group samples. *H19* expression was determined, and we observed a significant increase in samples from the cortex and the white matter, with a similar trend in the hippocampus; however, the differences were not significant. In samples extracted from the cerebellum, *H19* was neither up- nor downregulated (Figure 2B). *MALAT1* and *ANRIL* (Figure 2B) were determined to be significantly upregulated in the hippocampus and cerebellum, respectively. Differences in gene expression in the other brain regions were not significant. Interestingly, *TUG1* was the only lncRNA determined as significantly downregulated in the hippocampus, with a similar trend in the cortex (Appendix A). Gene expression in the hypoxia samples of *PANDA* (Appendix A) was neither up- nor downregulated compared with control samples. In addition, the melting curve analysis of *PANDA* displayed multiple peaks. Based on the inadequate results, we decided not to proceed with this lncRNA.

### 3.3. BDNF-AS Expression Was Significantly Increased after Hypoxia and Hyperoxic Exposure

Four of the lncRNAs were analyzed in samples from the cortex in all study arms, as defined in Figure 1, to investigate the potential difference in gene expression between normoxic and hyperoxic reoxygenation. *MALAT1* and *ANRIL* expressions were determined to be neither significantly increased nor decreased in all the groups. However, the intervention groups had more within-group variability than their respective control groups (Figure 3A). The results also indicated a similar trend between *MALAT1* and *ANRIL* expression, which was confirmed using a Pearson correlation plot, r = 0.8042 (Figure 3B). Analysis using qPCR revealed a significant increase in *BDNF-AS* expression in the hyperoxic groups compared with the control group (Figure 3C). Expression levels were not significantly changed between the three intervention groups. Although *H19* expression was slightly increased in the intervention groups, the difference relative to the control group and between groups was nonsignificant (Figure 3C). 

### 3.4. Analysis Using ddPCR Revealed Further Significant Increase in BDNF-AS Relative to the Time of Hyperoxic Reoxygenation

*BDNF-AS* and *H19* expressions were analyzed via ddPCR (Figure 3D). A significant difference between the medians of the BDNF-AS expression in samples from the intervention groups was demonstrated using a Kruskal–Wallis test. Post hoc analysis with Dunnett’s multiple-comparison test revealed that lncRNA expression was significantly increased in the Reox-100% (3′) group compared with the control group, and the level of significance was even higher in the Reox-100% (30′) group. Although the gene expression for the Reox-21% group was not determined to be significant, compared with the control group, the expression levels showed a trend toward an increase. Furthermore, the expression levels between the intervention groups were not significantly changed when compared with each other. Ordinary one-way ANOVA determined the means among *H19* in all the groups to be nonsignificant, and post hoc test Tukey’s multiple-comparison test revealed no further significance. However, *H19* in the Reox-100%(3′) group showed a tendency toward an increase compared with the control group (*p* = 0.0550). Data from qPCR and ddPCR were also compared using the Pearson correlation test and demonstrated a positive correlation of values for both *BDNF-AS* (r = 0.9781, *n* = 29, *p* < 0.0001) and *H19* (r = 0.9862, *n* = 29, *p* > 0.0001). Overall, there was a strong, positive correlation between the gene expression results analyzed using qPCR and ddPCR. 

## 4. Discussion

The “brave new world” of gene regulation by lncRNAs seems exceptional and fits well with the putative model of gene regulation proposed by Jacob and Monod [36] more than fifty years ago. Understanding the interaction of lncRNAs and their target genes in birth asphyxia may provide an opportunity to develop better and more relevant biomarkers of the severity of oxidative stress damage and novel therapeutic targets in the future. Numerous biological processes depend on low levels of ROS for normal cell activity; however, excessive ROS generation has deleterious effects on DNA, RNA, and proteins, thus causing oxidative stress and various diseases [37]. The antioxidant protective defense system is immature in preterm and term newborns, making the neonatal brain particularly susceptible to damage from oxidative stress and reperfusion/reoxygenation injury [38,39].

In this study, we investigated the changes and potential regulatory features of hypoxia-associated lncRNAs using a piglet model of perinatal asphyxia. To this end, our piglet model was facilitated with various exposure regimes, and samples were collected from diverse sections of the brain, including the cortex, the hippocampus, the white matter, and the cerebellum, to analyze protein-coding and nonprotein-coding RNAs using both qPCR and ddPCR.

Gene expression can be determined in a variety of ways, including Northern blot analysis, in situ hybridization, microarray technology, or RNA sequencing. QPCR is still considered the “gold standard” for quantitative gene expression because of its ubiquity, robustness, and relative ease of use. However, qPCR has its limitations for low-abundant genes, such as regulatory lncRNAs. Consequently, both qPCR and ddPCR with higher sensitivity and precision have been tested and applied.

### 4.1. The Effects of Hypoxic Exposure on Protein-Coding Genes Associated with Hypoxia

A panel of oxidative-stress-induced protein-coding genes was determined, including *TP53*, *TNFα*, *HIF1α*, *VEGFA*, and *BDNF.* The gatekeeper of the cell, *TP53*, is known to be activated in response to DNA damage and oxidative stress; however, our results did not reveal any differences between the control and Reox-21% groups, which might be explained by the cell type and the cellular condition-dependent activation of *TP53* [40,41]. Furthermore, *TNFα* expression was significantly upregulated only in samples from the cerebellum, and the within-group variability of expression was very high. *HIF1α* is a well-known regulator of the hypoxic response, promoting cell-type-specific responses [42]. Previous studies have reported various expression levels of *HIF1α* after hypoxia [43,44], and our results illustrate steady-state quantities of *HIF1α* in all of the brain regions. *HIF1α* is an early transcriptional response to acute hypoxia [45], suggesting that *HIF1α* levels had returned to normal by the time our samples were collected. *VEGFA* is part of the protective molecular machinery by replenishing blood flow, and this growth factor has been implicated in hypoxia–ischemia. Previous publications have reported an increase in *VEGFA* in samples subjected to hypoxia–ischemia [46,47], which is consistent with our results. *BDNF* is critical for brain development and plays an important role in stress responses and brain disorders [48]. Consequently, a decrease in its expression is associated with lower levels of protective mechanisms and an increase in hypoxic and oxidative stress [14]. However, our results demonstrated that *BDNF* levels were significantly increased in various brain regions after hypoxia and normoxic reoxygenation. This issue will be addressed in relation to *BDNF-AS* expression in more detail later.

### 4.2. Exposure to Hypoxia Resulted in Increased Expression of Various lncRNAs

Several lncRNAs have been reported to play a key role in brain injury, altering expression, and influencing the pathological outcome [14,32,33]. To date, hypoxia-associated lncRNAs have been mainly examined in the context of local hypoxia (e.g., tumor), but investigations of expression changes in other models of more global hypoxia are limited, and studies using a piglet model of perinatal asphyxia have not been carried out previously. A careful selection of hypoxia- and oxidative-stress-associated lncRNAs with potential regulatory features in perinatal asphyxia was performed (Table 1). *PANDA*, a lncRNA transcriptional target of *TP53* located at the p21 locus, has been reported to regulate apoptosis in response to DNA damage [25]. We confirmed the expression of the *PANDA* transcript in piglets, but the melt curve analysis displayed multiple peaks, indicating the presence of another target. Although our results did not reveal a significant alteration in expression, modification of the *PANDA* primers may allow a more accurate analysis of gene expression after hypoxic exposure. *TUG1* is an important lncRNA for retinal development, and hypoxia can cause extensive damage to the retina in humans and animals [49,50]. Consistent with other studies [27,51], our results demonstrate a significant decrease in the lncRNA in the hippocampus and a similar trend in the cortex, suggesting that *TUG1* is involved in the hypoxic response occurring in the brain. It has been reported that higher levels of oxidative stress and inflammation in the retina may enhance retinal neurodegeneration [52], but the exact mechanisms underlying perinatal asphyxia conditions have not been studied in detail [49]. In samples from the cortex and the cerebellum, *H19* levels were initially significantly increased in the Reox-21% group compared with the control group; however, in a combined analysis of all the groups, the changes were no longer significant. This is in agreement with the results reported by others, which highlight the variable expression of *H19* after hypoxic exposure [18,53,54,55,56]. Significantly higher expression of *MALAT1* and *ANRIL* was observed in various brain regions but only when analyzing the control versus the Reox-21% group, and not in the cortex, when analyzing all the groups. Nevertheless, we found considerable within-group variability compared with the control group, indicating that the lncRNAs were affected by various intervention regimens. In our analysis, we observed that the expression patterns of *MALAT1* and *ANRIL* in the different groups were quite similar, which was confirmed using a Pearson correlation plot (Figure 3B). In addition, other studies investigating the combined effect of *MALAT1* and *ANRIL* have reported a negative correlation with pathologic staging [57,58]. Therefore, we speculate that the attenuated expression levels of *ANRIL* and *MALAT1* in the groups exposed to 100% O_2_ (hyperoxia) may be negatively correlated with the severity of hyperoxic reoxygenation. 

### 4.3. BDNF-AS Expression Increases in a Time-Dependent Manner after Hyperoxic Reoxygenation

The protein-coding *BDNF* expression in humans is inversely correlated with the lncRNA *BDNF-AS* [15]. Consequently, the manipulation of *BDNF-AS*, e.g., under hypoxia and oxidative stress conditions, will have an effect on the protective mechanisms of *BDNF,* suggesting *BDNF-AS* as a possible target in the treatment of hypoxia–ischemia-induced brain damage [14]. In our study, both *BDNF-AS* and *BDNF* were significantly increased in various brain regions in the Reox-21% group compared with the control group (Figure 2). We observed a significant time-dependent increase in *BDNF-AS* expression, with 30 min of hyperoxic reoxygenation having a higher level of significance than 3 min of exposure. This is consistent with a previous study by Solberg et al. (2010), which demonstrated a dose-dependent decrease in *BDNF* expression when resuscitated with supplemental oxygen after hypoxic exposure [59]. We suggest that the time-dependent increase in *BDNF-AS* expression is due to the increased oxidative stress levels caused by hypoxia and hyperoxia, as shown by Qiao et al. (2020) in hippocampal neurons [14]. Accordingly, our combined results confirm the inverse correlation of *BDNF* and *BDNF-AS* expression after exposure to hypoxia. The achieved qPCR results obtained for *BDNF-AS* were confirmed using an independent technique, i.e., ddPCR (Figure 3D). 

### 4.4. Comparing Gene Expressions across Brain Regions 

This is one of the first studies to demonstrate the differential expression of lncRNAs under perinatal asphyxia conditions using a piglet model. The gene expression of protein-coding and non-coding hypoxia-associated genes was analyzed in different brain regions (Table 3). The diversity of expression between the various lncRNAs and brain regions may reflect their tissue specificity, as reported in other studies [12,60].

Our results provide important additional support for oxidative brain injury caused by hyperoxic reoxygenation in newborns. We propose that changes in the expression of *BDNF-AS* and other lncRNAs should be further investigated to gain a broader insight into their physiological and molecular mechanisms in birth asphyxia. 

### 4.5. Limitations of the Study

Pigs are very similar to humans on a genetic, brain developmental, physiological, and organic level. This evolutionary relationship further encompasses the genetic diversity among individuals, which is reflected in the variation in expression patterns [61,62]. The observed variations are methodologically challenging for molecular biology research, requiring a large number of individuals and study areas to keep the SD low; however, they more realistically reflect the actual situation we face in birth asphyxia in the clinic. 

In the present study, we investigated expression patterns in a perinatal asphyxia piglet model 9.5 h after hypoxic exposure. We would like to propose extending the current findings with longer time frames in the context of perinatal asphyxia using an additional model system such as mice. 

The identification of the cortex, the hippocampus, and the cerebellum is relatively straightforward, but collecting samples from the white matter is technically difficult, and contamination is more likely to occur.

## 5. Conclusions

Novel approaches are highly demanded to better understand the molecular mechanisms of hypoxic and oxidative stress in order to identify novel therapeutic targets for birth asphyxia in the future. To this end, we investigated the changes in the expression of different lncRNAs and target genes in various brain sections and after different intervention strategies. To the best of our knowledge, this is one of the first studies to investigate selected hypoxia- and oxidative-stress-associated lncRNAs, including *BDNF-AS*, *H19*, *ANRIL*, *TUG1*, and *PANDA*, in a perinatal asphyxia piglet model.

## Figures and Tables

**Figure 1 biology-12-00549-f001:**
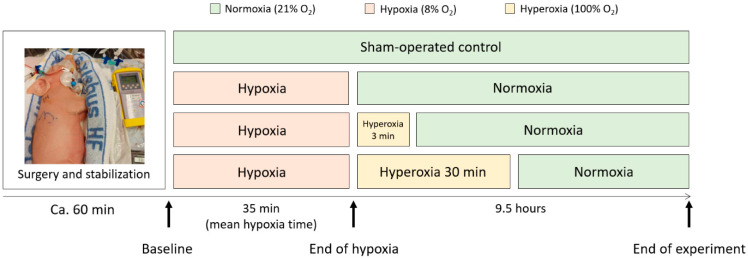
A schematic overview of the four study arms in the perinatal asphyxia piglet model. The piglets underwent anesthesia and surgery and were stabilized for 1 h before inducing hypoxia. The hypoxic mean time was 35 min, and the piglets were observed for 9.5 h after the end of hypoxia. The control group only received air (21% O_2_), while all treatment groups first received 8% O_2_ (hypoxia), followed by either 21% O_2_ (Reox-21%), 3 min of 100% O_2_ (Reox-100% (3′)) or 30 min of 100% O_2_ (Reox-100% (30′)).

**Figure 2 biology-12-00549-f002:**
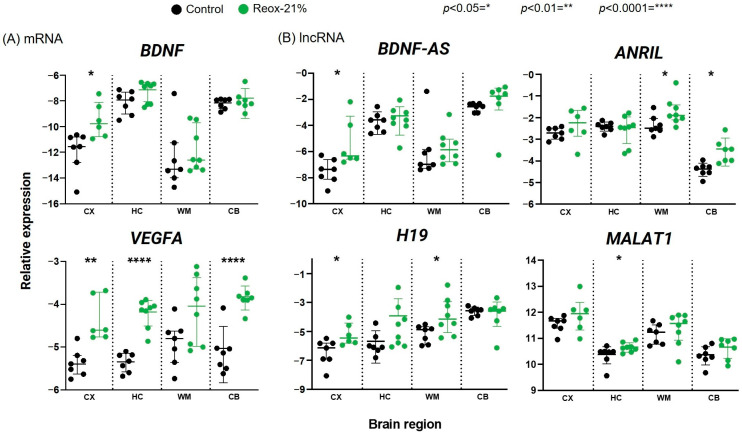
Differential expression of hypoxia- and oxidative-stress-regulated mRNAs (**A**) and lncRNAs (**B**) in the cortex (CX), hippocampus (HC), white matter (WM), and cerebellum (CB) of the brain in a perinatal asphyxia piglet model, analyzed using qPCR. The control group was exposed to 21% O_2_ during the entire procedure, while the Reox-21% group was exposed to hypoxia (8% O_2_) followed by normoxic reoxygenation (21% O_2_). Relative gene expressions (2^−∆Ct^) are shown on a log scale and expressed as either mean ± SD (*VEGFA*: HC, WM, CB; *BDNF*: CB; *BDNF-AS*: HC, CB; *H19*; *MALAT1*; *ANRIL*) or median ± IQR (*VEGFA*: CX; *BDNF*: CX, HC, WM; *BDNF-AS*: CX, WM).

**Figure 3 biology-12-00549-f003:**
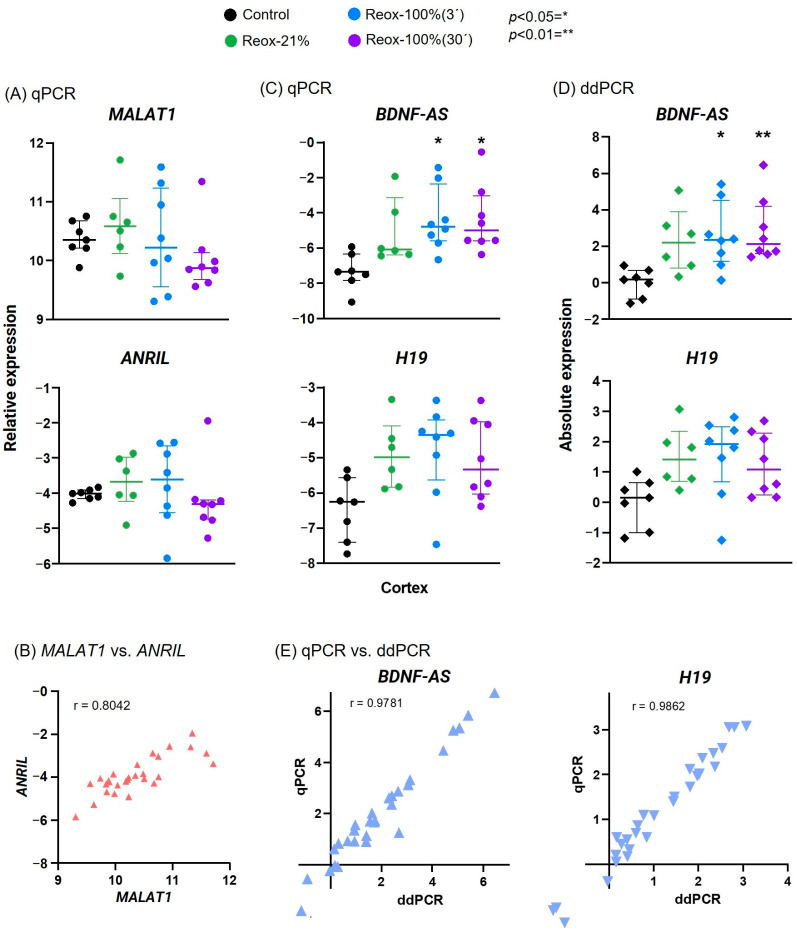
Hypoxia- and oxidative-stress-related lncRNAs after hypoxia and hyperoxic reoxygenation demonstrate expression differences between intervention groups. *MALAT1* and *ANRIL* expression in the cortex analyzed via qPCR was likewise affected by the hypoxia and hyperoxia (**A**) and verified using Pearson correlation plot (**B**). Hyperoxic reoxygenation analyzed via qPCR resulted in a significant increase in only *BDNF-AS*, although *H19* displayed a similar trend (**C**). When exposed to 30 min of hyperoxia, the increase in *BDNF-AS* expression was even more significant using ddPCR analysis (**D**). The expressions of *BDNF-AS* and *H19* measured via qPCR and ddPCR were compared and revealed a strong positive correlation (**E**). Reox-21%, hypoxia and normoxic reoxygenation; Reox-100%(3´), hypoxia and 3 min hyperoxia; Reox-100% (30´), hypoxia and 30 min of hyperoxia; control group only received 21% O_2_. Data are shown on a logarithmic scale as 2^−∆Ct^ and are expressed as either mean ± SD (qPCR: *H19*; ddPCR: *BDNF-AS*) or median ± IQR (qPCR: *BDNF-AS*; *MALAT1*; *ANRIL*; ddPCR: *H19*).

**Table 1 biology-12-00549-t001:** Summary of selected lncRNAs associated with hypoxia and oxidative stress, or induced by other stress factors.

lncRNA	Genomic Location	Associated mRNAs	Oxidative Stress	Hypoxia	Ischemia	Angiogenesis	Inflammation	Apoptosis	DNA Damage	References
** *BDNF-AS* **	11p14.1	*BDNF*	●	●						[14,15]
** *H19* **	11p15.5	*HIF1α*	●	●	●					[16,17,18]
** *MALAT1* **	11q13.1	*HIF1α*	●	●	●	●			●	[19,20,21,22]
** *ANRIL* **	9p21.3	*HIF1α*	●	●	●	●	●	●		[23,24]
** *PANDA* **	6p21.2	*P21*, *P53*							●	[25,26]
** *TUG1* **	22q12.2	*HIF1α*, *P53*	●	●	●			●		[27,28]

Abbreviations: *BDNF*, brain-derived neurotrophic factor; *HIF1α*, hypoxia-inducible factor α; *p53*, tumor suppressor gene p53; *BDNF-AS*, brain-derived neurotrophic factor antisense; *H19*, H19 imprinted maternally expressed transcript; *MALAT1*, metastasis-associated lung adenocarcinoma transcript 1; *ANRIL*, antisense non-coding RNA in the INK4 locus; *PANDA*, p21-associated ncRNA DNA-damage-activated; *TUG1*, taurine-upregulated gene 1.

**Table 2 biology-12-00549-t002:** Primer sequences of endogenous controls, mRNAs, and lncRNAs used for qPCR and ddPCR.

Working Name	Gene Name	(F/R)	Primer Sequence (5´-3´)	Tm °C
** *Endogenous* ** ** *controls* **				
*RPLP0* ^a^	Ribosomal protein, large, P0	F	ACAATGTGGGCTCCAAGCA	58.1
		R	CATCAGCACCACGGCTTTC	57.8
*TBP* ^b^	TATA-box binding protein	F	GACCATTGCACTTCGTGCC	58.7
		R	CTGGACTGTTCTTCACTCTTGGC	59.0
* **mRNAs** *				
*VEGFA*	Vascular endothelial growth factor A	F	ACGAAGTGGTGAAGTTCATGGA	57.5
		R	CACCAGGGTCTCGATTGGA	56.9
*BDNF*	Brain-derived neurotrophic factor	F	GTGACTGAAAAGTTCCACCAGGT	58.4
		R	CCTCGGACGTTGGCTTCTT	58.3
*HIF1α*	Hypoxia-inducible factor 1 subunit α	F	TGGCAGCAATGACACAGAAAC	58.4
		R	TGATTGAGTGCAGGGTCAGC	58.4
*P53*	tumor suppressor gene p53	F	AGCACTAAGCGAGCACTGCC	59.4
		R	CAGCTCTCGGAACATCTCGAA	59.2
*TNFα*	tumor necrosis factor α	F	CAAGGACTCAGATCATCGTCTCA	57.1
		R	CATACCCACTCGCCATTGGA	57.8
** *lncRNAs* **				
*BDNF-AS*	Brain-derived neurotrophic factor antisense	F	GGACAGAACAGTGGACTCTCAGACT	60.6
		R	CCCAGGTGTATGTTCTGCATCA	58.0
*H19*	Imprinted maternally expressed transcript	F	CCTGAACACTCTCGGCTGG	58.0
		R	GCTGGGTAGCACCATCTCTTG	58.4
*MALAT1*	Metastasis-associated lung adenocarcinoma transcript 1	F	CTGAAGCCTTTAGTCTTTTCCAGATG	59.8
		R	TTACTGGGTCTGGCTTCTCTGG	59.4
*ANRIL*	The antisense non-coding RNA in the INK4 locus	F	TGCTCTATCCGCCAATCAGG	59.8
		R	ACTCAGTGTCCAGATGTCGCAG	59.0
*PANDA*	P21-associated ncRNA DNA-damage-activated	F	GCTCTGATGTTTTCTTTGCCTTC	58.2
		R	ACATGACGAAGGGCCTTGTT	58.1
*TUG1*	Taurine-upregulated gene 1	F	CCCTGTCACTCCCAGATGTAGC	59.6
		R	AGCCAGGCTATGATCTGGAAGA	58.9

Abbreviations: F/R, forward/reverse primer; F, forward; R, reverse; Tm, melting temperature. Note: ^a^, endogenous control for mRNA; ^b^, endogenous control for lncRNA.

**Table 3 biology-12-00549-t003:** Summary of gene expression variations and significance across various brain regions in a perinatal asphyxia piglet model. Gene expression was measured in samples from the cortex, hippocampus, white matter, and cerebellum in piglets exposed to severe hypoxia (8% O_2_) and reoxygenated with 21% oxygen (Reox-21%), relative to the sham-operated control group (Figure 2 and Appendix A).

	Cortex	Hippocampus	White Matter	Cerebellum	
mRNA					Total
*HIF1α*	-	-	-	-	0/4
*VEGFA*	▲**	▲****	-	▲****	3/4
*BDNF*	▲*	▲	▲	-	3/4
*TP53*	-	-	-	-	0/4
*TNFα*	▲	-	-	▲*	2/4
**lncRNA**					
*BDNF-AS*	▲*	-	-	▲	2/4
*H19*	▲*	▲	▲*	-	3/4
*MALAT1*	-	▲*	-	-	1/4
*ANRIL*	-	-	-	▲*	1/4
*TUG1*	▼	▼*	*N/A*	*N/A*	2/2
*PANDA*	-	-	*N/A*	*N/A*	0/2
Total	6/11	5/11	2/9	4/9	

Abbreviations: *BDNF-AS*, brain-derived neurotrophic factor antisense; *H19*, H19 imprinted maternally expressed transcript; *MALAT1*, metastasis-associated lung adenocarcinoma transcript 1; *ANRIL*, antisense non-coding RNA in the INK4 Locus; *TUG1*, taurine-upregulated gene 1; *PANDA*, P21-associated ncRNA DNA-damage-activated; ▲, observed increase in expression; **▼**, observed decrease in expression; *p* < 0.05 = *; *p* < 0.01 = **; *p* < 0.0001 = ****.

## Data Availability

The data supporting the results of this study are available in DataverseNO at https://doi.org/10.18710/JIQKEU, accessed on 28 March 2023.

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
