# Peer review of "Long Non-Coding RNAs in Hypoxia and Oxidative Stress: Novel Insights Investigating a Piglet Model of Perinatal Asphyxia"

_biology, 2023, doi:10.3390/biology12040549_

Round 1

Reviewer 1 Report

The study explored several hypoxia-responsive lncRNAs in a porcine birth asphyxia model. The design is clear, and the analysis and results are straightforward. However, the whole study is quite simple, and therefore the purpose and significance of the study is relatively weak. There are some points that need to be modified.

  1. 1. Naming circRNAs using their host gene name is very confusing. If there are no general names, using "circ" plus the host gene name, such as circH19, is a better choice.

  2. 2. How many protein-coding genes were actually measured by qPCR? There were only two in Figure 1, but the author mentioned more in the description.

  3. 3. Please add a color code to label the control and treatment groups in Figure 2.
  4. 4. The numbering of figures was arranged disorderly. For example, in Result 3.3, the description starts from 3C, then 3A and 3B.
  5. 5. What is the point of ddPCR here? If it is just verification, I think it is not appropriate. Another type of analysis should be used here, such as Northern blot or in situ hybridization."

Author Response

Dear reviewer,

thank you very much for taking the effort to review our manuscript and for your helpful comments and suggestions, which we address point-by-point below.

  1. Naming circRNAs using their host gene name is very confusing. If there are no general names, using "circ" plus the host gene name, such as circH19, is a better choice.

There may have been a misunderstanding about the naming of the long non-coding RNAs (lncRNAs) studied and we apologize for the confusion. We agree with the reviewer that the identification of the these RNAs is challenging, due to a rich collection of synonymous names, e. g.; for ANRIL (Antisense noncoding RNA in the INK4 locus) synonymous names and descriptions include CDKN2B-AS1 (CDKN2B antisense RNA 1), p15AS (P15 antisense RNA), and ENSG00000240498. For the selected lncRNAs in our study, we used entirely general names, based on information extracted from the lncRNA database (https://rnacentral.org/), which is built on information from the NCBI and other genome browsers (Amaral PP et al., 2011). We present an overview of all included lncRNAs (Table 1.) together with an explanation of their abbreviated names and references to previous publications.

Based on our experience with the confusing synonym names, we feel it would be less appropriate to introduce a new label based on their loci or target gene, especially since lncRNAs may have multiple target genes and are only rarely positioned within a protein-coding gene.

  1. How many protein-coding genes were actually measured by qPCR? There were only two in Figure 1, but the author mentioned more in the description.

We are sorry, if the information was not communicated precisely enough. It is correct that the protein-coding genes VEGFA, BDNF, TP53, HIF1α, and TNFα were measured by qPCR in the cortex, hippocampus, white matter, and cerebellum, as stated in the Abstract, Materials and Methods, and the Results. The first paragraph in the results "3.1. Differences in the mRNA expression of target genes measured in various brain regions and after exposure to hypoxia and normoxic reoxygenation" describes the quantification of these mRNA target genes in more detail. For improved clarity, we have selected genes with significant changes (VEGFA and BDNF ) for illustration in the main manuscript (Figure 2.) and transferred figures exhibiting the expression patterns of all other mRNA genes to  Supplementary figure 1.

If the reviewer prefers that Figure 2 should be changed to include the "uninteresting" protein-coding genes, we are happy to change the figure; however, we would suggest leaving it.

  1. Please add a color code to label the control and treatment groups in Figure 2.

Thank you for pointing out this formal mistake. We are sorry that the color coding was lost during the transfer process and has now been added to the new version of the manuscript.

  1. The numbering of figures was arranged disorderly. For example, in Result 3.3, the description starts from 3C, then 3A and 3B.

The reviewer is completely right, the disorder of the figure sections in the result section 3.3 is unfortunate. The order has been corrected and is now as follows:

"Four of the lncRNAs were analyzed in samples from the cortex in (all study arms, as defined in Figure 1) to investigate the potential difference in gene expression between normoxic and hyperoxic reoxygenation. MALAT1 and ANRIL expressions were determined to be neither significantly increased nor decreased in all groups. However, the intervention groups had more within-group variability compared to their respective control groups (Figure 3A). The results also indicated a similar trend between MALAT1 and ANRIL expression, which was confirmed by a Pearson correlation plot, r=0.8042 (Figure 3B). Analysis by qPCR revealed a significant increase in BDNF-AS expression in the hyperoxic groups compared to the control group (Figure 3C). Expression levels were not significantly changed between the three intervention groups. Although H19 expression was slightly increased in the intervention groups, the difference relative to the control group and between groups was non-significant (Figure 3C)."

  1. What is the point of ddPCR here? If it is just verification, I think it is not appropriate. Another type of analysis should be used here, such as Northern blot or in situ hybridization."

We would like to emphasize that the rationale for performing ddPCR was multiple, confirmation of the qPCR was only one of several issues. QPCR has been specified as "the golden standard" for gene expression measurements. The technique is a very common, simple, and sufficient for most genes. Initial results showed us that some of the investigated lncRNAs are expressed at low levels, which is consistent with the literature on lncRNA expression. As a consequence, a more sensitive quantification method, ddPCR, was used. 

We agree with the reviewer that if confirmation studies were the main reason, other methods could have been performed, including microarrays, Northern blot or in situ hybridization. We have used Northern blot analysis extensively in previous studies (e.g.; Baumbusch et al., 2004); however, this technique it is less accurate than the "golden standard" qPCR or ddPCR. Microarray experiments or RNA sequencing are very costly, and in situ hybridization was not considered because tested antibodies were not at hand we wanted to investigate several lncRNAs in various tissues.

We would like to emphasize that this is a primary investigation of lncRNA in the context of perinatal asphyxia using a piglet model. We would like to add that, given the 10-days timeframe for the resubmission, we are afraid that additional wet-lab experiments would have been hard to perform. 

In order to meet the requests, we made two changes in the manuscript. In the result section, we changed the first sentence of the 3.4 paragraph to:

"BDNF-AS and H19 expression were analyzed using ddPCR (Figure 3D)."

In the discussion, we added a statement at the end of the first paragraph:

"Gene expression can be determined in a variety of ways, including Northern blot analysis, in situ hybridization, microarray technology, or RNA sequencing. QPCR is still considered as the "gold standard" for quantitative gene expression because of its ubiquity, robustness and relative ease of use. However, qPCR has its limitations for low-abundant genes, such as regulatory lncRNAs. Consequently, both qPCR and ddPCR with its higher sensitivity and precision have been tested and applied."

References:

Amaral PP et al., (2011) lncRNAdb: a reference database for long noncoding RNAs. Nucleic Acids Res.

Baumbusch LO et al., (2004) Levels of LEC1, FUS3, ABI3 and Em expression reveals no correlation with dormancy in Arabidopsis. J Exper. Bot.

Reviewer 2 Report

In this manuscript, Tune et al. describe changes in mRNA and lncRNA in a model of piglet perinatal asphyxia. The changes may provide groundwork for future investigations into understanding how organisms regulate hypoxia-driven responses. Overall, the manuscript is well-written and I suggest only minor changes.

Line 151: Was the omission of 2-mercaptoethanol chosen to prevent reduction of samples? Could the expression be effected by ex vivo oxidation? Please provide justification for this omission.

Supplemental Table 3. the superscript "d" is utilized in the footnotes, but I believe you meant "a," as this is what is actually found in the table. Also, MALAT1 has negative Ct values-how is this possible?

Line 215: How was the selection of lncRNA used for ddPCR made? Based on the qPCR results?

Figure 2: Please provide a legend to describe which is control and which is the reoxygenation group

Line 263: Referring to Figure 1 in parentheses here is confusing, as it can read as though you are presenting data in that figure. I would suggest stating "all study arms as defined in Figure 1," or something similar.

Throughout the manuscript, the term "tendency towards," or similar, is used. Please provide P-values for these data.

In the "limitations" section, it is suggested to use mouse models. Is this suggestion made solely because the sample size of mice can be much greater? It is confusing because pigs are a better model for human diseases.

Author Response

In this manuscript, Tune et al. describe changes in mRNA and lncRNA in a model of piglet perinatal asphyxia. The changes may provide groundwork for future investigations into understanding how organisms regulate hypoxia-driven responses. Overall, the manuscript is well-written and I suggest only minor changes.

Thank you very much for your encouraging words, which are highly appreciated.

Line 151: Was the omission of 2-mercaptoethanol chosen to prevent reduction of samples? Could the expression be effected by ex vivo oxidation? Please provide justification for this omission.

This is an interesting topic and we are happy to share our experience. For many years, many laboratories (including ours) have routinely used 2-mercaptoethanol for RNA extraction to prevent reduction. Due to its toxicity and odor, the chemical has to be handled with special care, making the extraction more complicated than a column- or magnetic-based techniques. A few years ago (2015), we came across an article published in Analytical Biochemistry by Mommaerts et al., which stated that 2-mercaptoethanol should be replaced by other methods. Consequently, we tested it in our laboratory with satisfying results. Differences in RNA yield and quality were negligible and as a result, we now omit 2-mercaptoethanol from our procedures.  

Supplemental Table 3. the superscript "d" is utilized in the footnotes, but I believe you meant "a," as this is what is actually found in the table. Also, MALAT1 has negative Ct values-how is this possible?

We thank the reviewer for pointing out this typographical error. "d" has been changed to "a" in the up-dated version of the supplementary.

We are sorry, if the calculation of the qPCR was not communicated clearly enough. The negative values for  MALAT1 are due to the formula that calculates the Ct-values against a reference value. Low values of the reference gene may lead to negative expression of the target lncRNAs.

"Gene expression studies were made using ViiA 7 RUO Software and absolute gene expression was calculated in Microsoft Excel using normalized Ct-values and the 2-∆Ct method before performing statistical analysis".

Line 215: How was the selection of lncRNA used for ddPCR made? Based on the qPCR results?

We agree that the procedure was not described precisely enough. There were several reasons why we decided to enlarge our studies by ddPCR. QPCR is a considered the "golden standard" for performing gene expression measurements. It is a quite common, robust and well-established method that is sufficient for many experimental settings, especially for the quantification of high abundance protein-coding genes. LncRNAs are often involved in regulatory functions and are therefore expressed at low levels, sometimes at the detection limit for qPCR. As a consequence, we wanted to try one of the most sensitive and precise methods. To clarify this strategy, we have added a statement at the end of the first paragraph in the discussion:

"Gene expression can be determined in a variety of ways, including Northern blot analysis, in situ hybridization, microarray technology, or RNA sequencing. QPCR is still considered as the "gold standard" for quantitative gene expression because of its ubiquity, robustness and relative ease of use. However, qPCR has its limitations for low-abundant genes, such as regulatory lncRNAs. Consequently, both qPCR and ddPCR with its higher sensitivity and precision have been tested and applied."

Further, in the result section, we changed the first sentence of the 3.4 paragraph to:

"BDNF-AS and H19 expression were analyzed using ddPCR (Figure 3D)."

We selected candidates based on their importance. As we chose an exploratory approach, we felt it would not make sense to verify lncRNAs that are not differentially expressed, such as PANDA). Other limitations included sample material, cost, and workload. Based on the significant levels of the primary experiments, we selected BDNF-AS, H19,  and ANRIL for advanced inquiries. We looked at the cortex region based on our experience and the fact that this is where the most dramatic changes occur and where in is most likely to exhibit dramatic expression differences. In addition, BDNF was known to reveal differential expression in the cortex. To this end, we think that regulation by BDNF-AS is a very interesting candidate for further and more detailed investigation.

Figure 2: Please provide a legend to describe which is control and which is the reoxygenation group

We are sorry about this mistake. The color coding was lost during the transferring process and it has now been added in the new version of the manuscript.

Line 263: Referring to Figure 1 in parentheses here is confusing, as it can read as though you are presenting data in that figure. I would suggest stating "all study arms as defined in Figure 1," or something similar.

We agree with the reviewer and have corrected the sentence as suggested.

Throughout the manuscript, the term "tendency towards," or similar, is used. Please provide P-values for these data.

This is a valuable suggestion. We provide P-values in the new version of the manuscript.

In the "limitations" section, it is suggested to use mouse models. Is this suggestion made solely because the sample size of mice can be much greater? It is confusing because pigs are a better model for human diseases.

We totally agree with the reviewer that pigs are a better model for human diseases, as mentioned in our statement in the limitations section of the study:

"Pigs are very similar to humans on a genetic, brain developmental, physiological, and organic level. This evolutionary relationship further encompasses the genetic diversity among individuals, which is reflected in the variation of expression patterns (61, 62)."

Working with a pig model incorporates several challenges. Of course, pig experiments are costly and many experimental set-ups are too small, not taking into account the high variability of biological parameters in pigs. However, the main problem when working with perinatal asphyxia is different. For practical and animal welfare reasons, we are not allowed to keep the animals overnight. We have to terminate the experiments after 9,5 hours at the latest. We know from patients that both, short- and long-term effects of hyper-and hyperoxia may accumulate. This makes it impossible to study long-term effects, which is a major drawback of our model that could be overcome by rodent models. We have expressed this suggestion in the following limitation statement:

"The present study has investigated expression patterns in a perinatal asphyxia piglet model 9.5 hours after hypoxic exposure. We would like to propose to extend the current findings with longer time frames in the context of perinatal asphyxia using an additional model system such as mice."

We have changed the order of our statement to clarify this issue.

References:

Mommaerts et al., 2015, Replacing β-mercaptoethanol in RNA extractions. Analytical Biochemistry.

Reviewer 3 Report

Overall the data presented in the manuscript embodies valuable information regarding the role of long non-coding RNAs in regulating hypoxia and oxidative stress. This is a very important issue because of the frequency of perinatal asphyxia and the number of deaths in neonates. The authors have planned the experiments well and presented the data in a way that will help the scientific community to study birth asphyxia, a relatively preventable cause of death in newborn infants.

There are a few spelling/grammatical errors that need to be corrected.  For instance, the sentence “The observed variations are methodological challenging for molecular biology research…” can be modified as “The observed variations are methodologically challenging for molecular biology research…”.

Kindly explain why when we look at the figures 2B and 3C, the expression of BDNF-AS in the cortex has a significant difference (p<0.05) between control and Reox-21% groups in fig 2B whereas there is no difference between the groups in 3C.

Author Response

Overall the data presented in the manuscript embodies valuable information regarding the role of long non-coding RNAs in regulating hypoxia and oxidative stress. This is a very important issue because of the frequency of perinatal asphyxia and the number of deaths in neonates. The authors have planned the experiments well and presented the data in a way that will help the scientific community to study birth asphyxia, a relatively preventable cause of death in newborn infants.

Thank you very much for your pleasant comments. We share your opinion and with your kind support, we will be able to proceed with the publication process.

There are a few spelling/grammatical errors that need to be corrected. For instance, the sentence “The observed variations are methodological challenging for molecular biology research…” can be modified as “The observed variations are methodologically challenging for molecular biology research…”.

Thank you for highlighting this mistake and for your suggestions. Several spelling and grammatical mistakes were detected in a detailed language correction step and have now been corrected in the new version of the manuscript.

Kindly explain why when we look at the figures 2B and 3C, the expression of BDNF-AS in the cortex has a significant difference (p<0.05) between control and Reox-21% groups in fig 2B whereas there is no difference between the groups in 3C.

We apologize that the experiments and the statistical approaches were not communicated clearly enough.

Briefly, the difference is based on both, the experimental design and the applied statistics. More specifically, the results in Figure 2 show the changes in lncRNAs and mRNAs expression changes between the Reox-21% versus control group in various regions of the brain. In Figure 3, all lncRNA expression measurements are performed in the cortex purely, focusing on the differences between the various intervention groups and the control.

Initially, we analyzed the difference in expression of BDNF-AS and other lncRNA and mRNA across various tissues using a t-test (Figure 2).

In the follow-up experiments (Figure 3), we looked at differences in the cortex solely, but compared diverse intervention groups. Several  statistical models were applied to analyze the results in Figure 3: Kruskal-Wallis test and  Dunn’s multiple comparisons test. While post-hoc tests, such as Dunn's multiple comparisons test, are effective in controlling family-wise error rate, they reduce the individual significance level for each pair-wise comparison. The greater the number of comparisons, the lower the significance level applied resulted in a lower statistical power. Thus, a study with lower power is less likely to detect group mean differences that may exist in the population.

In conclusion, the significance level of BDNF-AS expression observed in Figure 2B may be due to the application of a different statistical model that was more sensitive. However, in figure 3C, the lower statistical power associated with the use of Dunn's multiple comparisons test may have limited our ability to detect group mean differences in BDNF-AS expression in the cortex.

Round 2

Reviewer 1 Report

The auther have largely improved the presentation of results and the expression is much more clearer now.